# Impact of influenza vaccination on amoxicillin prescriptions in older adults: A retrospective cohort study using primary care data

**Lauren R. Rodgers**[1]*, **Adam J. Streeter**[2], **Nan Lin**[3], **Willie Hamilton**[1], **William E. Henley**[1]

**1** Institute of Health Research, University of Exeter Medical School, Exeter, United Kingdom, **2** Medical Statistics, Faculty of Health: Medicine, Dentistry & Human Sciences, University of Plymouth, Plymouth, United Kingdom, **3** Department of Mathematics, Physics and Electrical Engineering, Northumbria University, Newcastle upon Tyne, United Kingdom

* L.R.Rodgers@exeter.ac.uk

## Abstract

### Background

Bacterial infections of the upper and lower respiratory tract are a frequent complication of influenza and contribute to the widespread use of antibiotics. Influenza vaccination may help reduce both appropriate and inappropriate prescribing of antibiotics. Electronic health records provide a rich source of information for assessing secondary effects of influenza vaccination.

### Methods

We conducted a retrospective study to estimate effects of influenza vaccine on antibiotic (amoxicillin) prescription in the elderly based on data from the Clinical Practice Research Datalink. The introduction of UK policy to recommend the influenza vaccine to older adults in 2000 led to a substantial increase in uptake, creating a natural experiment. Of 259,753 eligible patients that were unvaccinated in 1999 and aged≥65y by January 2000, 88,519 patients received influenza vaccination in 2000. These were propensity score matched 1:1 to unvaccinated patients. Time-to-amoxicillin was analysed using the Prior Event Rate Ratio (PERR) Pairwise method to address bias from time-invariant measured and unmeasured confounders. A simulation study and negative control outcome were used to help strengthen the validity of results.

### Results

Compared to unvaccinated patients, those from the vaccinated group were more likely to be prescribed amoxicillin in the year prior to vaccination: hazard ratio (HR) 1.90 (95% confidence interval 1.83, 1.98). Following vaccination, the vaccinated group were again more likely to be prescribed amoxicillin, HR 1.64 (1.58,1.71). After adjusting for prior differences between the two groups using PERR Pairwise, overall vaccine effectiveness was 0.86 (0.81, 0.92). Additional analyses suggested that provided data meet the PERR assumptions, these estimates were robust.

**Data Availability Statement:** The data used in this study are owned by the Clinical Practice Research Datalink (www.cprd.com). Researchers can apply for data access from CPRD and submit a study

protocol to the Independent Scientific Advisory Committee for approval, using the information outlined in the Methods section of the manuscript. The authors had no special access privileges to the data that future researchers would not have.

**Funding:** The author(s) received no specific funding for this work.

**Competing interests:** WH has a personal long-term shareholding in Glaxo Wellcome, who manufacture an influenza vaccine. WEH has previously received funding from IQVIA. This does not alter our adherence to PLOS ONE policies on sharing data and materials.

## Conclusions

Once differences between groups were taken into account, influenza vaccine had a beneficial effect, lowering the frequency of amoxicillin prescribing in the vaccinated group. Ensuring successful implementation of national programmes of vaccinating older adults against influenza may help contribute to reducing antibiotic resistance.

## Introduction

The influenza vaccine has been shown to be effective in reducing the incidence of secondary respiratory infections [1–6]. As respiratory illnesses are difficult to diagnose precisely in the early stages, patients with respiratory symptoms, particularly the elderly, are often prescribed antibiotics. Antibiotic prescription may serve as a proxy for influenza presenting as a respiratory infection (inappropriate prescribing) or a possible bacterial infection complicating influenza (appropriate prescribing) [1, 7, 8]. The routine prescribing of antibiotics for acute respiratory infections risks the evolution of antimicrobial resistance through their overuse [9–12]. Therefore, antibiotic use as an outcome is increasing in importance, e.g. in an ongoing trial of influenza vaccination in patients with diabetes [13]. A recent review [14] found that antibiotic prescriptions related to respiratory illness were reduced in children receiving the influenza vaccination.

Influenza vaccination effectiveness (IVE) in the elderly has been an ongoing subject of debate, with previous studies suggesting that much older patients gain less protection [15] due to immunosenescence [4, 16, 17] and confounding due to frailty [18, 19]. The source of uncertainty arises because there are relatively few randomised controlled trials (RCTs) in older patients and much of the evidence comes from observational studies. The trials that have been conducted have tended to consider a younger healthier subset of adults ≥65yrs and often lack power to draw conclusions from more representative samples from this age group [4, 20], or older subsets [5, 21]. In a cost-effectiveness evaluation of the vaccination in the elderly, Newall et al [22] stratify their analysis into five-year groups; although no details on group differences were given, overall the vaccination was found to be cost-effective. Electronic healthcare records (EHR) are an important source of information for studies of vaccine efficacy in real-world settings. However, bias has been identified in observational studies using these data [5, 23–25] Influenza is one vaccine in particular where confounding may influence efficacy estimates. Some studies have proposed using the period between influenza seasons to assess the bias between exposure groups, although it is uncertain whether the bias is the same as that during influenza circulation [26, 27]. A further complication arises when considering the degree to which the effect of vaccination carries over into the next influenza season [28–30]. This is difficult to assess as IVE can vary depending on how well matched the vaccine is to the current virus strain [4, 30]. Dependent on the type of confounding, IVE can be under or overestimated [29, 31, 32]. Possible disease covariates have been identified [3, 7, 26] but the potential remains for residual confounding, arising from unmeasured or unidentified covariates, to impact on estimates of IVE from EHR data.

In this study, we assess the effectiveness of the influenza vaccine in reducing antibiotic prescription for acute respiratory infections amongst adults aged ≥65yrs using EHR data from the Clinical Practice Research Datalink (CPRD). In 2000 the UK introduced a policy to offer the influenza vaccination to all adults aged ≥65yrs. Health Protection England reported a vaccination level of 46% in 1999, 65% in 2000 and subsequent increase to current levels at >70%

[33]; recruiting a large cohort of never-before-vaccinated patients created the basis for a natural experiment to assess the impact of influenza vaccination on antibiotic prescription [7]. We apply the prior event rate ratio (PERR) Pairwise method [34, 35] which can reduce bias in estimates from EHR data, and has been used in a previous study of influenza vaccination effectiveness [36]. We look at age sub-groups as part of this study to evaluate any difference due to age.

## Material and methods

### Study cohort

CPRD [37] contains primary care records from 45 million patients registered at General Practices in the UK. Patients who had reached the age of 65y by the year 2000 were extracted from CPRD. 259,753 patients met the inclusion criteria for this study (Fig 1, S1 Fig). The PERR methodology requires that patients should never have had the treatment prior to the study period. In the case of influenza vaccination, this may not be the case. To avoid any residual vaccination effect, we selected patients who were vaccination free for at least two years before the study period. 88,519 had an influenza vaccination in the 2000 season. Vaccinated patients were matched to controls based on a propensity score (PS) model using 1:1 nearest neighbour matching. Matching was based on disease covariates, selected according to clinical risk groups [38] and the Quality Outcomes Framework [39]; a proxy for patient health using a binary variable for ≥12 GP consultations [4] between 01/09/1999–31/08/2000; smoking status, age, gender, ethnicity and region were also included. Table 1 lists all variables (age-dementia) our cohort were matched on.

### Ethics approval

Approval for the study was granted by the CPRD Independent Scientific Advisory Committee (ISAC 14_159R2). Patient data received from CPRD is fully anonymised.

### Outcomes

There are no outcome measures in primary care data that explicitly identify viral influenza infection; laboratory confirmed influenza is not well recorded in primary care and all-cause mortality is not considered useful [4–6]. The main effect of influenza is respiratory; there is evidence of a reduction in respiratory illness due to the influenza vaccination [1–3]. As respiratory illnesses are difficult to diagnose precisely in early stages, patients, particularly the elderly, are prescribed antibiotics. Antibiotic prescription may serve as a proxy for influenza presenting as a respiratory infection or possible bacterial infection complicating influenza [1, 7, 8] Amoxicillin is frequently prescribed for respiratory illness [11] and antibiotic prescription is recorded reliably in CPRD data [40]. The outcome measure for this study was time to

---

**Inclusion criteria**
Aged over 65 in 2000
Continuously registered with GP between September 1998 and May 2001
Alive on 01/09/2000

**Exclusion criteria**
Have a vaccination in the two years prior to 2000 influenza season
Multiple vaccinations in 2000 influenza season
Death within 14 days of vaccination

---

**Fig 1. Inclusion and exclusion criteria for study.**

**Table 1. Demographic characteristics of each group.** Mean (SD) or percentage reported.

| | Vaccinated Group (n = 88519) | Control Group (n = 88519) |
|---|---|---|
| **Prescribed amoxicillin in prior period** | 7.5% | 4.0% |
| **Prescribed amoxicillin in study period** | 7.1% | 4.4% |
| **Dies in study period** | 1.3% | 1.2% |
| **Age Mean(SD)** | 73.3 (6.7) | 73.3 (6.6) |
| **Gender (% male)** | 44.6% | 45.7% |
| **Ethnicity White** | 74.4% | 74.7% |
| Unknown | 24.1% | 23.8% |
| Non-White | 1.6% | 1.5% |
| **Smoking Current** | 47.6% | 47.4% |
| Ex-smoker | 10.0% | 10.4% |
| Non-smoker | 42.1% | 42.1% |
| **More than 12 GP consultations** | 9.6% | 8.7% |
| **Asthma** | 13.8% | 13.1% |
| **Mental Health** | 0.9% | 0.8% |
| **Epilepsy** | 3.1% | 3.0% |
| **Thyroid** | 5.0% | 4.9% |
| **Hypertension** | 28.6% | 28.1% |
| **Stroke** | 5.3% | 5.1% |
| **Chronic Heart Disease** | 11.9% | 11.3% |
| **COPD** | 2.7% | 2.5% |
| **Heart Failure** | 2.8% | 2.6% |
| **Cancer** | 5.2% | 5.0% |
| **Depression** | 9.6% | 9.4% |
| **Diabetes** | 1.8% | 1.6% |
| **Arterial Fibrillation** | 3.1% | 2.9% |
| **Dementia** | 0.6% | 0.6% |

amoxicillin prescription. Relevant amoxicillin codes were identified by a pharmacist. September was used as the start of the influenza season; the prior period was defined between 01/09/1999–30/04/2000 and the study period 01/09/2000–30/04/2001; Fig 2. The adjusted start date for each vaccinated patient was the vaccination date plus 14 days to allow for antibodies to

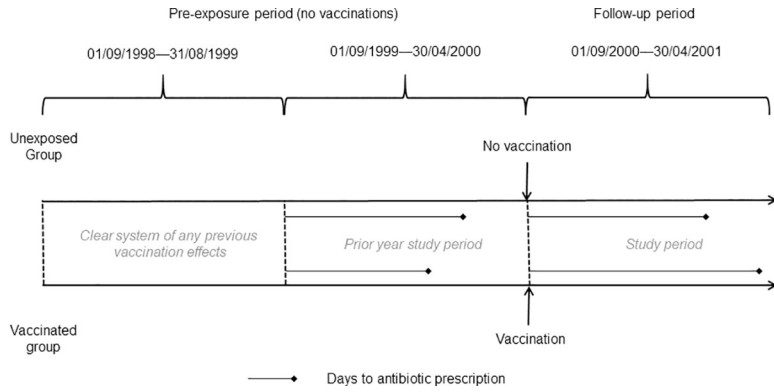

**Fig 2. Study design schematic.**

reach protective levels [33]. The cohort was followed in the study period from adjusted start date until they either received antibiotics, died or reached the end of the study period. The start date in the prior period was the adjusted vaccination date minus one year; patients were followed until they either received amoxicillin prescription or reached the end of the period. Control patients were assigned start dates for follow-up based on the vaccination dates of vaccine recipients, to whom they had been matched.

## Statistical analysis

To tackle bias arising from unmeasured confounding in EHR data, several methods have been proposed [41, 42] with some previously applied for estimating influenza vaccine effects [6, 27, 43]. One promising method is the prior event rate ratio (PERR), and an extension of this method, PERR Pairwise [34, 35]. The PERR approach involves fitting a model for event outcomes after the start of treatment (the study period) and adjusting treatment estimates based on a model for the same outcome in a prior period. By selecting a cohort of vaccinated and unvaccinated patients that have never received the treatment before the study period, the hazard ratio (HR) for the prior period should reflect the effect of confounders, measured and unmeasured, independent of vaccination status. Simulation studies [34–36, 44] have found that bias in treatment effect estimates in the presence of unmeasured confounders can be reduced using this methodology.

To assess IVE in a real-world setting, we apply Cox proportional hazards (PH) models, PERR and PERR Pairwise to CPRD data. Cox (PH) models were fitted to the outcome in the prior period and the study period. Hazard ratios (HRs) for the risk of antibiotic prescription in the vaccinated group vs the unvaccinated group are calculated for each time period, $HR_{prior}$ and $HR_{study}$. Any difference between the vaccinated and unvaccinated groups in the prior period (i.e. if the 95% confidence interval for $HR_{prior}$ does not include 1) is assumed to reflect unmeasured confounding.

The PERR methodology adjusts the $HR_{study}$ using the estimate of the underlying difference between the two groups identified in the prior period, $HR_{prior}$, as follows:

$$HR_{PERR} = \frac{HR_{Study}}{HR_{Prior}}$$

The extension to this method, PERR Pairwise, uses a paired Cox regression and reduces bias found in the original formulation [34, 35]. Mathematical details of the statistical models, and R code, are reported elsewhere [34, 35, 45]. After matching on the propensity score, vaccination status was the only covariate in the models.

CRPD data were extracted using Stata v15.0 and analysed using R v3.0.2.

## Sensitivity analyses

Previous work has proposed that negative control outcomes (NCO) [45] and simulations should accompany analyses using observational data [46]. We used incidence of oedema as an NCO to test the robustness of applying PERR Pairwise to produce real-world estimates of IVE; this is unrelated to amoxicillin prescription and we would expect to find no difference between the two groups (S1 File). A simulation study (S2 File) was designed to replicate the features of our motivating data with an unmeasured confounder influencing outcome. We explored the impact on IVE estimates if the model assumptions for the PERR methodology were violated. In particular, we simulated scenarios in which vaccinated and control groups could have different characteristics, confounders were allowed to be time-varying (comorbidities over the

prior and study periods) and there was heterogeneity in response to vaccination, i.e. no sub-groups (S2 File, S1 Table).

## Results

### Study cohort

Of those over 65yrs in our data, 38% (n = 515,580) received the influenza vaccination in 1999 and 56% in 2000 (n = 514,291); this is reasonably comparable to the increase in coverage reported by Public Health England. In our study population, the vaccinated group received a greater number of amoxicillin prescriptions in both periods than the control group, Table 1. There is a difference in the number of prescriptions in the vaccination-free prior period, 7.5% for the group to who went on to be vaccinated vs 4.0% for the control group. This indicates the presence of a pre-existing difference in the health of the exposure groups or a difference in healthcare-seeking behaviour; those who were vaccinated were more likely to be prescribed amoxicillin. Prescriptions in the vaccinated group decreased to 7.1% in the study period, whereas prescriptions in the control group increased to 4.4%.

### Influenza vaccine effectiveness

The results from the propensity score-adjusted Cox PH model for the prior period showed a difference in the hazards between the vaccinated and control groups. The bias was not fully removed by matching on the propensity scores (which were derived from the measured con-founders alone) (for the ≥ 65y cohort: HR 1.90 95% CI [1.83, 1.98]; Fig 3). The HR from the PS-adjusted Cox model for the study period was above 1, corresponding to an implausible harmful effect of vaccination (for the ≥ 65y cohort: HR 1.64 95% CI [1.58, 1.71]). However, accounting for group-differences in outcomes in the prior period with PERR and PERR

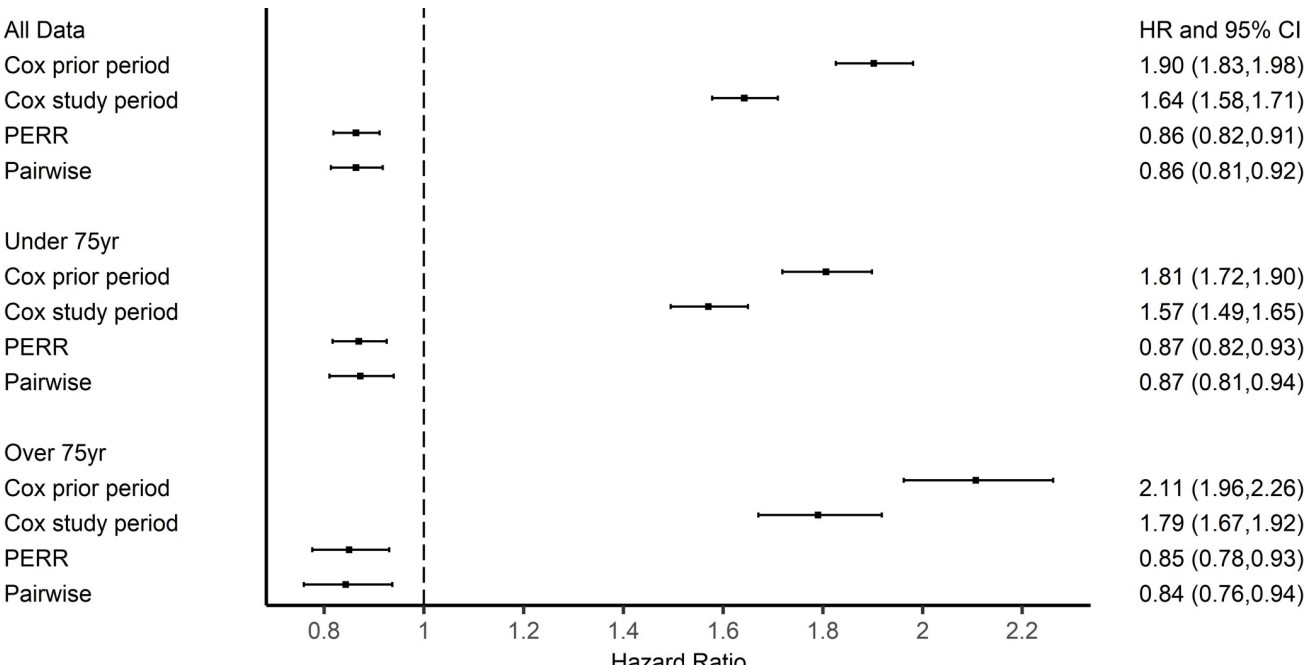

**Fig 3. Results of influenza vaccination effectiveness in 2000 season.**

Pairwise models, gave a protective effect estimate for vaccination, with HRs 0.86 (0.81,0.92) and 0.86 (0.81, 0.92) respectively.

## Age

The subgroups based on age showed similar results. More prescriptions were issued to the vaccinated groups in the prior period, 7.5% in <75yrs and 7.6% in ≥75yrs in the vaccinated group and 4.3% in <75yrs and 3.6% in ≥75yrs for the control group. Similarly, in the study period, rates of prescriptions were 7.0% and 7.1% for <75yrs and ≥75yrs respectively in the vaccinated group and 4.5% and 4.0% respectively in the control group. We again found evidence of a difference between the groups before the study period and Cox PH models for the study periods (Fig 3) indicating a greater likelihood of antibiotic prescription in the vaccinated group in both age groups in the prior period. The PERR Pairwise adjustment found a beneficial response for the <75y age group, PERR Pairwise 0.87 (0.81,0.94). A beneficial effect of the influenza vaccination for ≥75yrs was also found, HR 0.84 (0.76,0.94).

## Sensitivity analyses

After adjusting for the prior period with the PERR methods, the artefactual association between oedema, our NCO, and influenza vaccination was eliminated, HR 1.05 (0.95,1.16) and 1.06 (0.95,1.19) for PERR and Pairwise, respectively (S1 File, S2 Fig). In the simulation study (S2 File, S3 and S4 Figs, S1 and S2 Tables), we tested the robustness of estimates to violations of the PERR methodology assumptions (S1 Table). We found that bias would be induced if there was a change between the prior and study periods or if a proportion of the vaccinated group did not respond to the vaccination (immunosenescence). The study provided no evidence for a difference in IVE between age groups suggesting immunosenescence was not an important factor; thus, providing there were no substantial change over time within patients, consistent with our NCO analysis, the estimates using PERR are robust. The PERR and Pairwise results in the simulation studies were similar and only Pairwise is presented in S2 File.

## Discussion

In this study, we found evidence of a beneficial effect of influenza vaccination in reducing antibiotic prescribing in EHRs, by taking a robust approach to the problem of confounding and applying methods to adjust/mitigate for confounding bias. In the year that all ≥65yrs were first entitled to a free vaccination, having accounted for the prior period, vaccination reduced the risk of being prescribed antibiotics; those vaccinated had a 14% (8–19%) lower risk of being prescribed the commonest antibiotic used for respiratory infection than those who were not vaccinated. Those who were vaccinated in 2000 were different to the control group who were not vaccinated: in both periods the vaccinated group were at greater risk of being prescribed antibiotics; the HR for vacinees vs controls in the prior period being 1.90 (1.83,1.98) and 1.64 in the period after vaccination (1.58,1.71). Examining the period post-vaccination in isolation, resulted in a spurious estimate of IVE corresponding to a harmful effect of being vaccinated–in practice, patients due to accept influenza vaccination are more likely to be prescribed antibiotics for respiratory infections because of an increased likelihood of healthcare-seeking behaviour.

### Strengths and limitations

Given the strain of the influenza virus can vary from year to year, it is possible that not only will vaccination effectiveness vary but the influence of the confounders may also vary. Bias

may be introduced into estimates from the PERR methods when the influence of unmeasured confounders on response is different in the prior and study periods. The PERR methodology also requires unmeasured confounders to be time invariant. Some advice should be taken prior to applying the PERR methodology about how comparable the prior and study periods are. In this case consultation with a virologist about the strains of influenza and their impact on health in the two seasons should inform study design. In this study, patients identified with a vaccine-free prior period could have a history of exposure to the vaccine thus potentially contaminating the prior period and reducing the performance of the method in adjusting for bias. However, a strength of the PERR methodology is that it allowed us to adjust for the prior incidence of antibiotic prescriptions and greatly reduce the imbalance between the two groups without full knowledge of the underlying confounders. A naïve model could use prior use of antibiotics as a covariate in the study model but this would not be as effective in reducing bias due to confounding as the PERR adjustment [45]. Our NCO outcome, oedema, showed bias in estimates using prior and study periods only. Applying the PERR methodology removed a false association between influenza vaccination and oedema.

The simulation study showed where we could 'trust' our estimates of vaccine efficacy. Our simulations are novel in that they utilise biologically plausible baseline hazards and adding a small competing risk. In scenario 1, the PERR methods were robust to differences in the distributions of the continuous confounder in the vaccinated and control groups, provided they remained the same within each group over the two periods, whereas we found high bias in the Cox model in comparable settings.

In scenario 2, the confounding variable which influenced the groups changed between periods. We found biased estimates in this scenario, however confidence intervals still contained the true effect. Assuming our vaccinated and control groups did not change significantly in their underlying health between the two time periods, then our PERR estimates were valid. We attempted to capture health seeking behaviour that may contribute to variation in outcomes using the number of GP visits in one year as a covariate. This approach may fail to pick up where health seeking behaviour has changed over time and further work is required to investigate how best to incorporate these time dependent effects within the PERR framework. The simulation study scenario 3 showed that, if there were a subgroup who did not respond to vaccination, any analysis would be biased. We found no difference in response to vaccination due to age in our cohort which suggested that immunosenescence did not impact on the results.

Our study concentrated on amoxicillin prescription as an outcome to study reduction in antibiotic prescription following influenza vaccination. Incidence of influenza is poorly recorded in EHR data and proxies related to respiratory illness, such as amoxicillin prescription, provide a feasible alternative. Amoxicillin can be used for many different types of bacterial infections but we chose to use a sensitive outcome definition without restriction to amoxicillin prescriptions where there was an associated clinical code for respiratory illness. We note that some patients may have been allergic to penicillin and unable to receive amoxicillin. Amoxicillin is the most commonly prescribed antibiotic for respiratory illness [11]; but further work could explore other antibiotics.

## Comparison to other literature

Klugman and Black [14] reviewed studies of antibiotic resistance after influenza vaccination. One efficacy study, reporting the effect on antibiotic prescriptions found 13.2% fewer prescriptions in the six months after vaccination [47]. Another study found 36% fewer prescriptions in the vaccine group in Asia Pacific, 59% in Central America and 71% in Europe as well as a reduction in antibiotic prescriptions across five seasons [48]. However, both of these studies

were in young children. An observational study in the UK [49] found a reduction of 14.5% in amoxicillin prescription during the influenza season which is a similar figure to our study, but again this was a study in children. Kwong et al [7] looked at the effect of the Canadian introduction of an influenza programme in 2000 and found a risk Ratio 0.36 (0.26–0.49). This 64% reduction in risk of antibiotic prescription covers a population from 6 months upwards. Our study found a reduction of 14% (8–19%) in antibiotic prescriptions in those ≥65yrs. This matches the positive effect of influenza vaccination on antibiotic prescription seen in other studies [14]: however, no other studies have looked at this in our aged population.

Our results did not find a diminished protective effect with age; 13% (6–19%) reduction in risk of prescriptions for <75 yrs and 16% (6–24%) in those ≥75yrs. Previous work has commented on the lack of evidence for effectiveness in much older people [4, 5, 15, 16, 26], and while we found a beneficial effect in ≥75yrs here, we would caution against interpreting this as definitive evidence; age may be too simple a subgrouping. Functional status, location and social connectivity have been identified as possible subgroups [2, 29, 50], but such data are not available in CPRD. Further work should explore more complex subgroups.

## Conclusions

We have found evidence, using EHR data, that vaccinating patients aged at least 65 years against influenza reduces prescriptions of a common antibiotic used to treat respiratory infections. This effect still holds when looking at a subgroup of over 75years. Antibiotic resistance is a growing healthcare problem, the consequences of which include longer healthcare stays and more expensive healthcare costs. Our findings suggest that improving uptake of influenza vaccination in older patients can contribute to strategies to reduce antibiotic resistance.

## Supporting information

**S1 Checklist.**
(DOCX)

**S1 Fig. Flow chart of patients entering the influenza vaccination study.**
(PDF)

**S2 Fig. Negative control example.**
(PDF)

**S3 Fig. Simulation study scenario 1.** 95% Confidence interval coverage of assessing change in continuous confounder distribution (an interval containing zero covers the true treatment effect).
(PDF)

**S4 Fig. Simulation study scenario 2.** 95% Confidence interval coverage of assessing change in continuous confounder distribution (an interval containing zero covers the true treatment effect).
(PDF)

**S5 Fig. Simulation study scenario 3.** Percentage labels indicate the percentage of patient who do not respond to the vaccination e.g. 50% half of the treated patients do not respond to the vaccine. 95% Confidence interval coverage of assessing change in continuous confounder distribution (an interval containing zero covers the true treatment effect).
(PDF)

**S1 Table. Description of simulation study scenarios.**
(DOCX)

**S2 Table. Results of simulation studies.** Figures are relative bias (%) of treatment effect across $\beta_{trt}$ from analysis of data simulated from parameters under the three scenarios.
(DOCX)

**S1 File. Negative control outcome.**
(DOCX)

**S2 File. Simulation study.**
(DOCX)

## Acknowledgments

The Age UK Project team at the University of Exeter for Read codes for the disease covariates.

## Author Contributions

**Conceptualization:** Willie Hamilton, William E. Henley.

**Data curation:** Lauren R. Rodgers.

**Formal analysis:** Lauren R. Rodgers.

**Methodology:** Lauren R. Rodgers, Adam J. Streeter, Nan Lin, William E. Henley.

**Supervision:** William E. Henley.

**Visualization:** Lauren R. Rodgers.

**Writing – original draft:** Lauren R. Rodgers.

**Writing – review & editing:** Lauren R. Rodgers, Adam J. Streeter, Nan Lin, Willie Hamilton, William E. Henley.

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
