## [Decision Letter · Decision Letter 0]

24 Nov 2020

PONE-D-20-35449

Impact of influenza vaccination on amoxicillin prescriptions in older adults: A retrospective cohort study using primary care data

PLOS ONE

Dear Dr. Rodgers,

Thank you for submitting your manuscript to PLOS ONE. After careful consideration, we feel that it has merit but does not fully meet PLOS ONE’s publication criteria as it currently stands. Therefore, we invite you to submit a revised version of the manuscript that addresses the points raised during the review process.

We look forward to receiving your revised manuscript.

Kind regards,

Sreeram V. Ramagopalan

Academic Editor

PLOS ONE

Journal Requirements:

"I have read the journal's policy and the authors of this manuscript have the following competing interests: WH has a personal long-term shareholding in Glaxo Wellcome, who manufacture an influenza vaccine. WEH has previously received funding from IQVIA."

Reviewers' comments:

Reviewer's Responses to Questions

**Comments to the Author**

1. Is the manuscript technically sound, and do the data support the conclusions?

Reviewer #1: Yes

2. Has the statistical analysis been performed appropriately and rigorously? 

Reviewer #1: Yes

3. Have the authors made all data underlying the findings in their manuscript fully available?

Reviewer #1: Yes

4. Is the manuscript presented in an intelligible fashion and written in standard English?

Reviewer #1: Yes

5. Review Comments to the Author

Reviewer #1: Your outcome of interest (Amoxicillin use) is potentially too vague. Amoxicillin can be used for many different types of bacterial infections. It is not clear why you did not use diagnoses codes for respiratory infections.

Would also have used time-varying cox regression to assess vaccination status. You potentially have introduced immortal time into the vaccinated cohort.

Would be nice to see the covariates you matched on. Did you control for senility? You mention health seeking behavior. Can you think if any proxies that you could use to account for health seeking behavior?

6. PLOS authors have the option to publish the peer review history of their article (what does this mean?). If published, this will include your full peer review and any attached files.

Reviewer #1: No

---

## [Author Response · Author response to Decision Letter 0]

12 Jan 2021

Response to editor comments

 The paper has been updated to conform to PLOS ONE’s requirements.

We have added “Patient data received from CPRD is fully anonymised.” to our ethics statement section.

"I have read the journal's policy and the authors of this manuscript have the following competing interests: WH has a personal long-term shareholding in Glaxo Wellcome, who manufacture an influenza vaccine. WEH has previously received funding from IQVIA."

 We have updated our competing interests statement:

WH has a personal long-term shareholding in Glaxo Wellcome, who manufacture an influenza vaccine. WEH has previously received funding from IQVIA. This does not alter our adherence to PLOS ONE policies on sharing data and materials. Data are owned by the Clinical Practice Research Datalink (www.cprd.com). Researchers can apply for data access from CPRD and submit a study protocol to the Independent Scientific Advisory Committee for approval.

Thank you.

The data used in this study are owned by the Clinical Practice Research Datalink (www.cprd.com). Researchers can apply for data access from CPRD and submit a study protocol to the Independent Scientific Advisory Committee for approval, using the information outlined in the Methods section of the manuscript. The authors had no special access privileges to the data that future researchers would not have.

The ethics statement has been moved to the end of the Study Cohort section of Materials and Methods 

Response to reviewer comments

1. Your outcome of interest (Amoxicillin use) is potentially too vague. Amoxicillin can be used for many different types of bacterial infections. It is not clear why you did not use diagnoses codes for respiratory infections.

We thank the reviewer for this comment. There is difficulty is selecting an appropriate outcome measure in a study of influenza vaccination using electronic health records; outcomes must be related to the effect of the treatment and well recorded. As noted in the outcomes section of the paper, laboratory confirmed influenza is not well recorded in primary care and all-cause mortality is not considered a useful outcome. Influenza attacks the respiratory system and there is evidence of a reduction in respiratory illness due to influenza vaccination [Gross et al, Nichol et al, Monto et al]. As respiratory illnesses are difficult to diagnose precisely in early stages, patients, particularly the elderly, are often prescribed antibiotics. Antibiotic prescription is well-recorded in routine health records and may serve as a proxy for influenza presenting as a respiratory infection or possible bacterial infection complicating influenza[Kwong et al, Nichol et al, Harper et al]. Our chosen outcome in this study was prescription of amoxicillin, the first choice antibiotic recommended by the National Institute for Health and Clinical Excellence (NICE) for low or moderate severity community acquired pneumonia. Although we could have further qualified our outcome by only selecting amoxicillin prescriptions where there was an associated clinical code for respiratory illness, we chose to use a more sensitive definition in this exploratory study to maximise available sample size. Our choice of within-patient design (based on the PERR method) should ensure that we remove the effect of any differences in levels of amoxicillin prescription for non-respiratory infections between vaccinated and unvaccinated groups. Text has been added to the Material and Methods Outcomes section to clarify our choice of outcome measure. Relevant amoxicillin codes were identified by a pharmacist. We acknowledge the limitations of our study in paragraph 3 of Strengths and Limitations section of the Discussion and agree further work should investigate this outcome measure further.

 Gross PA, Hermogenes AW, Sacks HS, Lau J, Levandowski RA. The efficacy of influenza vaccine in elderly persons. A meta-analysis and review of the literature. Ann Intern Med 1995;123:518–27

Nichol KL, Treanor JJ. Vaccines for seasonal and pandemic influenza. J Infect Dis 2006;194 Suppl:S111–8. doi:10.1086/507544.

Monto AS, Hornbuckle K, Ohmit SE. Influenza vaccine effectiveness among elderly nursing home residents: a cohort study. Am J Epidemiol 2001;154:155–60.

Kwong JC, Maaten S, Upshur RE, Patrick DM, Marra F. The effect of universal influenza immunization on antibiotic prescriptions: an ecological study. Clin Infect Dis 2009;49:750–6. doi:10.1086/605087

Harper SA, Fukuda K, Uyeki TM, Cox NJ, Bridges CB. Prevention and control of influenza. Recommendations of the Advisory Committee on Immunization Practices (ACIP). MMWR Recomm Rep 2005;54:1–40.

2. Would also have used time-varying cox regression to assess vaccination status. You potentially have introduced immortal time into the vaccinated cohort.

Although immortal time bias could be a potential problem with the PERR methodology as, by design, patients must be alive at the start of the study period, this issue has been addressed by previous authors including Tannen et al (2009) in response to a comment on their original paper. As with their original study design, our patients in both exposure groups had a predefined time interval to meet entry criteria for the study (no vaccination for two years prior to the study period). There was a large pool of unvaccinated patients (n=171274) who were matched to vaccinated patients based on the characteristics in Table 1. Their start date for the study period was effectively random as it was allocated by their matched vaccinated counterparts. Immortal time bias is introduced when the period of “immortality” before exposed patients receive treatment is either misclassified with regards to treatment status or excluded from the analysis. Neither of these issues apply here and Tannen et al point out that immortal time bias should not be an issue due to the careful choice of design. Furthermore, we have previously conducted a simulation study that showed the Pairwise PERR method is unbiased in the presence of differential case fatality because of the within-patient nature of the comparisons (Lin and Henley, 2016). We also note that the PERR framework set out by Lin and Henley (2016) allows for flexible modelling including use of multiple time-varying covariates and effects (Section 7.1). The version of the Pairwise PERR model that we use in this study does indeed allow for the time-varying effect of vaccination status. 

Tannen Richard L, Weiner Mark G, Xie Dawei. Use of primary care electronic medical record database in drug efficacy research on cardiovascular outcomes: comparison of database and randomised controlled trial findings BMJ 2009; 338 :b81

Lin NX, Henley WE. Prior event rate ratio adjustment for hidden confounding in observational studies of treatment effectiveness: a pairwise Cox likelihood approach. Stat Med. 2016 Dec 10;35(28):5149–69.

3. Would be nice to see the covariates you matched on. Did you control for senility?

Subjects were matched on the covariates in Table 1. This has been clarified in Material and Methods Study Cohort section. We have matched on dementia to control for senility (Table 1).

4. You mention health seeking behaviour. Can you think if any proxies that you could use to account for health seeking behaviour?

We have tried to capture health seeking behaviour through including the number of GP visits over the 12 months of the prior period as a covariate. We have used the number of consultations within one year as a snap shot of health seeking behaviour and assumed this to be representative during our study. However, we acknowledge that a change in consultations over this time period may be indicative of changing health. A limitation of current PERR methodology is the assumption that any unmeasured confounders are time invariant. We have added to the discussion this limitation (paragraph two of strengths and limitations) and highlighted the need for further work.

---

## [Editor Report · Decision Letter 1]

15 Jan 2021

Impact of influenza vaccination on amoxicillin prescriptions in older adults: A retrospective cohort study using primary care data

PONE-D-20-35449R1

Dear Dr. Rodgers,

We’re pleased to inform you that your manuscript has been judged scientifically suitable for publication and will be formally accepted for publication once it meets all outstanding technical requirements.

Kind regards,

Sreeram V. Ramagopalan

Academic Editor

PLOS ONE
---

## [Editor Report · Acceptance letter]

21 Jan 2021

PONE-D-20-35449R1 

Impact of influenza vaccination on amoxicillin prescriptions in older adults: A retrospective cohort study using primary care data 

Dear Dr. Rodgers:

I'm pleased to inform you that your manuscript has been deemed suitable for publication in PLOS ONE. Congratulations! Your manuscript is now with our production department. 

Kind regards, 

on behalf of

Dr. Sreeram V. Ramagopalan 

Academic Editor

PLOS ONE